# Factor Influences for Diagnosis and Vaccination of Avian Infectious Bronchitis Virus (Gammacoronavirus) in Chickens

**DOI:** 10.3390/vetsci8030047

**Published:** 2021-03-16

**Authors:** Md. Safiul Alam Bhuiyan, Zarina Amin, Ag Muhammad Sagaf Abu Bakar, Suryani Saallah, Noor Hydayaty Md. Yusuf, Sharifudin Md. Shaarani, Shafiquzzaman Siddiquee

**Affiliations:** 1Biotechnology Research Institute, Universiti Malaysia Sabah, Jln UMS, Kota Kinabalu 88400, Sabah, Malaysia; dr.safiulalambhuiyan@gmail.com (M.S.A.B.); zamin@ums.edu.my (Z.A.); suryani@ums.edu.my (S.S.); hydayaty@ums.edu.my (N.H.M.Y.); 2Jabatan Perkhidmatan Veterinar Sabah, Makamal Diagnosa Veterinar Kota Kinabalu, Peti Surat No 59, Tanjung Aru 89457, Sabah, Malaysia; agmuhdsagaf.abubakar@sabah.gov.my; 3Food Biotechnology Program, Faculty of Science and Technology, Universiti Sains Islam Malaysia, Bandar Baru Nilai, Nilai 71800, Negeri, Malaysia; sharifudinms@usim.edu.my

**Keywords:** vaccination, diagnosis, cross-reactions of serotypes, challenge and immunization

## Abstract

Infectious bronchitis virus (IBV) is a major economic problem in commercial chicken farms with acute multiple-system infection, especially in respiratory and urogenital systems. A live-attenuated and killed vaccine is currently immunized to control IBV infection; however, repeated outbreaks occur in both unvaccinated and vaccinated birds due to the choice of inadequate vaccine candidates and continuous emergence of novel infectious bronchitis (IB) variants and failure of vaccination. However, similar clinical signs were shown in different respiratory diseases that are essential to improving the diagnostic assay to detect IBV infections. Various risk factors involved in the failure of IB vaccination, such as various routes of application of vaccination, the interval between vaccinations, and challenge with various possible immunosuppression of birds are reviewed. The review article also highlights and updates factors affecting the diagnosis of IBV disease in the poultry industry with differential diagnosis to find the nature of infections compared with non-IBV diseases. Therefore, it is essential to monitor the common reasons for failed IBV vaccinations with preventive action, and proper diagnostic facilities for identifying the infective stage, leading to earlier control and reduced economic losses from IBV disease.

## 1. Introduction

Infectious bronchitis virus (IBV) is an acute and highly contagious respiratory pathogen in chickens. In general, coronaviruses are classified into four groups (Alpha, Beta, Gama, and Delta) according to antigenic cross-reactivity and nucleotide sequence analysis [1]. IBV belongs to genus Gammacoronavirus with positive-sense single-stranded RNA (+) ssRNA genome; its gene organization is 5′UTR-1a/1ab-S-3a-3b-E-M-5a-5b-N-3′UTR [2,3]. Moreover, betacoronaviruses are included human coronaviruses such as SARS-CoV (severe acute respiratory syndrome coronavirus) and SARS-CoV-2 (COVID-19) [4,5]. The length of the IBV genome is approximately 27.6 Kb, and the virion is encircling 5′ and 3′ untranslated regions (UTRs) with a poly (A) tail [6], the functional and nonfunctional genes are shown in Figure 1. Nine functional genes were encoded in the IBV genome, four genes were involved as main structural proteins, and another five genes are known as nonstructural proteins (Nsps). Most IBV genotypes and serotypes are closely associated with the vaccine strains or variants that are very distinct based on geographical areas circulating each lineage (GI) on complete nucleotide sequences of S1 (spike) gene [7,8]. Moreover, the distribution and multiplicity of these IBV genotypes vary with different geographical areas [9]. The major IBV serotypes are characterized in Massachusetts in the USA, 4/91 (793B or CR88) in the UK, D274 (D207, D212, or D1466, D3896, and D3128) in Europe, QX-like reported in China, H120 strain in the Netherlands with several variants introducing local and regional region by transmissions [10,11]. Different types of serotype or genotype of IBV failed to cross-protection against multiple genotypes due to continuous evolution of IBV [12,13,14,15]. Immunization with multiple serotype vaccines can be given 90% protection against different types of IBV infection instead of homologous vaccines (i.e., “protectotype concept”) [16,17,18].

IBV infection has major economic impact on the poultry industry due to mutable tissue tropism and the continuous emergence of various IBV serotypes or genotypes in different geographical regions. Because of its degree of severity and high contagiousness, mortality can reach up to 10%–60% over four to six weeks of broiler age during acute infection with a secondary infection [19,20]. There are no appropriate policies to control or prevent IBV infections without correct vaccination, especially live-attenuated and killed vaccines manufactured from local strains or serotype-specific immunity [21]. Farmers normally follow rotational vaccination programs for controlling IBV; however, the incidence of disease has become a repeated occurrence with local variants [22]. Vaccination is the most significant method for the prevention and control of IBV in the field. Previous reports demonstrated that live vaccines commonly stimulated both local protection and systemic immunity, but inactivated vaccines provide prolonged immunity after inducing with live-attenuated vaccine [23]. The killed IBV vaccine is applied either single or combined of two or more serotypes in bivalent vaccines [24]. Maternally derived antibodies (MDA) of progeny chicks are received from vaccinated breeder hens with inactivated vaccines as a substitute for live vaccine [25,26].

The failure of the vaccine is a consequence of the incapability of birds to produce a satisfactory immune response after vaccination [27]. Various factors are associated with accurate IBV vaccinations, such as the prospect of long-term immunity, the selection of maximal virulent serotypes, and the timing of applications according to flocks requiring revaccination [28]. Farmers blame the vaccine’s lack of effectiveness for failure to immunize their flocks. More than 50% of vaccination failures were recognized in vaccinated flocks due to the improper application of vaccine. Moreover, the increased risk of incorrect administration of vaccines, and cold-chain maintenance and storage of vaccine quality are important issues for vaccine failure leading to outbreaks of IBV in vaccinated farms [29]. Proper attention to those common factors responsible for failure vaccines provides reduced IBV costs and problems in poultry farms. The most significant part of IBV diagnosis identified genotype selection by molecular assay for correct vaccine candidates. Differential diagnosis needs to be established due to similar clinical signs exposed by various respiratory pathogens as required for proper testing to avoid misdiagnosis. This is the reason why diagnosis needs to be set up through the accumulation of combined serology data matched with clinical signs and isolation of the virus for the interpretation of results. Most poultry producers routinely use ELISA testing for the early diagnosis, challenge of infection, and failure of vaccination of IBV. Even though many factors are considered to increase, or abnormal IBV titers as necessary to check the timing of vaccination, seroconversion before and after vaccination, history of field challenge, geometric mean titer (GMT), and coefficient of variation (CV%), are necessary to meet the standards with the purpose of final diagnosis of IBV challenge or good vaccination [30,31].

The aim of this review article is mainly to discuss advanced current challenges related to proper diagnostic assay with evidence of vaccination failure. We describe various factors influenced by IBV vaccination, various routes of vaccination, postvaccine challenge, possible immunosuppression, and differential diagnosis to explore future research to elucidate the immune-pathogenesis response of multiple types of IBV.

## 2. Factors Influencing IBV Diagnosis

Few assays are established for infectious bronchitis (IB) diagnosis for the identification of viruses from current infections or specific antibody responses from post-viral infection. The selection of the best assay for the successful detection of IBV is difficult and confusing due to various factors, as discussed below.

### 2.1. Clinical Signs

The clinical signs of IBV are similar to those in other respiratory infections except for a few unique lesions that are part of strain variation and difficult to differentiate from other respiratory viral infections. Although clinical signs are matched according to the manual, a serology study is not ever matched according to autopsy lesions. The serology titer is occasionally given higher trends, but it cannot match with clinical signs; therefore, IBV confirmation is problematic. Thus, conformation diagnosis is based on the accumulation of data from farm records, including egg quality with production graph, mortality, seroconversion or antibody levels, vaccination report, and blood serology, which are clinical signs that concurrently match symptoms of IBV chickens through PM lesions [31,32,33]. Sample collection for IBV isolation is best when immediately achieved, as clinical signs are obvious. PCR on reverse-transcribed RNA is an effective method for detecting IBV in active infections. Some studies suggested that nutritional deficiencies are an important consideration for detecting IBV [34,35].

### 2.2. Vaccination

Vaccination is the main influence for IBV identification, as it requires the infected IBV-vaccinated strain and other virulent field strains. For unvaccinated flocks, the field challenge can simply be detected by RT-PCR, RT-qPCR, or a positive IBV titer after a few weeks. Therefore, there is great difficulty in identifying a field challenge or active infection in vaccinated flocks due to antibody circulation in the bloodstream. Nowadays, vaccine strains can be distinguished from field strains through the differentiating infected from vaccinated animals (DIVA) technique using inactivated oil emulsion vaccine that can be detected by simple neutralization test, enzyme linked immunosorbent assays (ELISA) and indirect immunofluorescence test [36,37,38].

### 2.3. Assay Selection

The selection of diagnostic techniques is important for proper diagnosis related to sample collection and sampling numbers. Advanced ELISA tests can distinguish antibodies owing to vaccination from those from infection. The concentration of antibody titer tends to be abnormally increased after 2–4 weeks post infection; the titer gradually decreased slowly in subsequent weeks [39]. Subsequently, the next blood sample should be collected after 2 weeks, but no later than 4 weeks in the same infected houses [39] because the titer is delayed in the highest level due to the immune-system’s need for time to build up immunity. The highest concentration of virus was found in infected tissue during the first 3–5 days post infection; there is a need for the selection of molecular assays and viral isolation instead of serology monitoring. Common molecular practices are currently applied for IBV diagnosis, particularly in RT-PCR, but the IBV isolate is more challenging, time-consuming, and exclusive from infected chicken due to requiring several passages in embryonated chicken eggs until the virus replicates. Alternatively, primers need to be frequently updated to apply in the molecular assay due to various types of variants that can escape identification because of frequent mutations occurring in the primer binding site.

### 2.4. Serology

The mean titer provides information on the antibody response of tested birds within a flock after vaccination. Mean titers should reach a predictable range with uniform distribution for normal flocks and unintentionally higher titers for diseased flocks. A study suggested that antibody patterns should not be breached over vaccinal immunity because of field infections [40]. Post infection, the geometric mean titer (GMT) should be abnormally increased when titer level is at least 2 times higher than the expected level after vaccination or at least 2 times the mean titer level before the infection [31,32]. Mean titer is sometimes gradually elevated after vaccination; however, no clinical signs were found with normal production, indicating good protection and better response to vaccine [31].

On the basis of general parameters, a percentage coefficient of variation (CV) that is maintained below 40% after inactivated vaccinations and less than 60% in live vaccines suggests good vaccination uniformity. When CV is less than 20%, the level of titer shows good uniformity, but previous exposure of disease or good vaccination is specified [32]. Therefore, it is necessary to cross-match with mortality, clinical signs, and production records for the final confirmation of IBV. A good assessment of CV percentage is significantly below the expected level after vaccination or significantly below levels in earlier infection cases. Consequently, CV% can reach 40%–70% after a single shot of live IBV vaccine, but below 30% is specified as a suspected challenge. Conversely, after vaccinating with high immunogenic IBV variants, CV percentage is commonly less than 45% [32]. In this situation, a series of multiple live vaccinations are initially used to boost the birds for 100% seroconversion before killed vaccination, which is a major principle for influence on the persistence of titers rather than measure of percent CV [41,42]. In serology detection systems, ELISA tests are commonly used to determine the trend analysis of protective antibody titers by good vaccination or unexpected serology, which is caused by virulent serotype or variant of IBV. However, ELISA results can be difficult to identify seroconversion against specific serotypes or variants except serotype-specific ELISA, e.g., Ark or Mass, Conn or Del 072 [43]. For this reason, the HI (Hemagglutination inhibition) test may contribute more valuable serotype-specific evidence, if birds are exposed to one diverse IB serotype. Apart from that, IBV titers tend to be less stable during production compared with titers of infectious bursal disease (IBD) and Newcastle disease (NDV).

### 2.5. Organ Selection and Sample Quality

The upper respiratory tract (UTI) is the principal site of IBV replication; subsequently, viremia causes viral multiplication to disseminate to other tissue through blood circulation [44,45,46]. IBV can be persistent in the long term in the cecal tonsils and kidneys due to the continuous cross-infection of infected or immune flocks [44,47]. In acute cases, clinical signs are identified in the respiratory tract, which is the preferred spot for pooling samples. Furthermore, kidney, cecal tonsil, and cloaca samples are selected in chronic infections in vaccinated chickens in the laying stage, whereas small amounts of the virus are predictable in the respiratory tract [48]. The pooling organ should be refrigerated faster to preserve viral viability for further molecular diagnosis. If there is no freezing or refrigerating facility, samples should be kept in 50% glycerin for several days.

### 2.6. Immunization and Challenge during Infection

Serological monitoring can be directly helpful for the assessment of the immune status of birds, viral challenge, and the duration of viral infection [19]. Different types of commercial ELISA kits were calculated using different cut-offs and mathematical formulations to convert the ELISA result. The most common practice in farming to build up immunity for controlling IBV infection is by live-attenuated vaccines followed by inactivated vaccines given the normal titer level for good protection against IBV [49]. Two phases of serum samples are essential to compare for proper diagnosis, as first samples are taken from the early infection, and the second stage of samples are taken after 3–4 weeks. The delay of the first sampling can prevent the detection of seroconversion [50]. For unvaccinated flocks, an IBV field challenge can simply be to confirm by identifying positive IBV titers in serology. For vaccinated IBV flocks, the presence of a field challenge is more difficult due to vaccinated birds showing a certain limit of titer level, which is required to compare with abnormal titers. However, unexpectedly increasing titers were significantly higher than expected vaccination titers with a lower CV%, indicating the existence of a field challenge [51]. IBV challenge normally shows respiratory signs while examining abnormally high IBV titers after 6–8 weeks post infection. In the case of breeder chicken, the normal instruction to calculate challenged birds is the mean titer of post infection being at least a 2 or 3 times the link to be expected from vaccination titers (baseline), as shown in Figure 2A,B [31,32]. In the case of broilers, an abnormal IBV titer is normally shown in the challenge, while the mean titer post-infection should be over 3–4 K (Idexx std) after a live vaccination link to normal serology, as presented in Figure 2C,D, respectively. In this review article, all serological data are evaluated on the basis of field trials collected from Turan breeder and broiler farms, Sabah, Malaysia, using commercial ELISA Kit (IDEXX, Westbrook, ME, USA). In the ELISA test, diagnosis should be established through expected vaccination titers depending on the vaccination program, the geographical distribution of IBV, outbreak history, specific vaccines, and type of inspected bird. Those respective factors are associated with the combination of periodic flock reports to determine whether serological results are normal or abnormal. Many factors influence levels of antibodies, particularly the strain of the virus, and some factors related to the chicken (e.g., breed, type, age, immune status, and nutrition), vaccine (e.g., levels of attenuation, dosage, live or killed vaccine, and storage conditions), and vaccination schedule (e.g., proper schedule, preparation, and route of administration) [52].

### 2.7. Genetics and Immunosuppression

Genetic traits have great influence on resistance and susceptibility to IBV. Some experimental studies showed the different genetic traits, mortality, viral growth, and variable histopathology found in different breeding lines after IBV immunization [53]. However, different detection methods have some dissimilarity because of the difference in susceptibility among chicken lines. Some immunosuppressive infections are associated with exacerbating the IBV disease, such as infectious bursal disease (IBD), Marek’s disease (MD), inclusion body hepatitis (IBH), and chicken infectious anemia (CIA) potentially suppress immune status at an early age, which can disturb the acquired vaccinal immunity of IBV by stimulated lymphocyte depletion from the bursa and thymus, causing higher resistance to challenge [54]. Major damage is affected by subclinical infection, making it difficult to understand immunosuppressive agents [55]. Laboratory diagnostic methods for confirming IBV infection concern other respiratory viruses that require continuous serology monitoring to identify the actual source of current infections. Similarly, cyclosporine is classified as a T-cell immunosuppressive drug that can significantly reduce the immunity of birds and is highly stimulated in IBV infections [56].

## 3. Vaccination Factor Influence

The level and duration of vaccination responses rely on many factors that need to be fully considered during vaccination. It is not sufficient to simply describe a specific IBV vaccine that can protect against infection in a particular situation, since regulatory requirements are commonly challenged.

### 3.1. Stress and Vaccination

According to Dantzer [57], stress is “the nonspecific response to environmental stimulation, the result of extreme demands placed on the physiological and behavioral experiences of birds to adaptation”. All kinds of stress can decrease a bird’s immune system, related to various factors such as temperature fluctuation, high relative humidity, poor nutrition, parasitic infestation, and disease condition. Vaccinations should occur in healthy birds with good conditions to avoid stress. The adverse results of stress on the immune defense system are the main effects of the glucocorticosteroid hormone; therefore, inhibiting the synthesis of prostaglandin and leukotriene mediators of inflammation, resulting in the suppression of the immune defense system. A study suggested evidence of a negative impression between stress and antibody response to vaccination, which is the most superficial with thymus-related vaccines and was calculated a long time after vaccination [58] A period of chronic stress was found to significantly decrease the total number of T-helper, T-lymphocyte, and B-lymphocyte cells due to continuous raising of corticosterone hormone levels that exceeded the allostatic load [58,59,60]. However, prolonged stress in young chickens leads to the premature regression of lymphatic or immune organs, for example, the thymus, bursa of Fabricius, and spleen [61]. Combinations of effects on the body metabolism and immune system of stressed birds do not tolerate vaccine acceptance; it is ultimately unclear if they properly confer protection.

### 3.2. Methods for Protection Studies

A protection study is not a simple method for application, but several factors were considered during designing such studies [62]. Different methods were used for performing in vivo protection studies using the overall concept of vaccinating and challenging groups to assess whether a vaccine protects against a particular challenge of a local strain. Arvidson et al. [63] established a different approach of viral clearance test from the lungs with particular vaccination and challenge via the trachea. This study allowed for determining an index of protection score that was more than 50% as a protection level, while 90% of the rings had regular activity [64]. The titer of a vaccine prohibited the replication of a challenge virus in the lungs evaluated by titrating lung homogenates in an in vitro ciliostasis test [62]. Studies used lung samples rather than the trachea to minimize the risk of detecting residual effects due to lung tissue supporting IBV growth to a similar level as that in the trachea. Ciliary protection stimulated by IBV vaccination programs against three groups of virulent IBV strain challenges on day 21 post vaccine are shown in Figure 3 [65]. Experimental studies reported that killed vaccines applied without priming with live-attenuated vaccines stimulated lower systemic immune responses, as no more local protection, especially tracheal ciliostasis, leads to systemic infection affected by virulent IBV serotypes [66].

### 3.3. Vaccine and Vaccination Program

Prior to vaccination, a program needs to be revised according to epidemiological study, the prevalence of a virulent virus, data of sporadic outbreaks, accurate diagnosis, single or combined IBV vaccine, vaccine potency, and monitoring poultry health systems. Immunization is commonly dependent on the challenge of establishing their effectiveness against multiple variants of viruses; a specific time is favorable for the multiplication of viruses, live-attenuated or killed vaccines, and monitoring the titer level or baseline of determination. Essential information details whether bird immunization is protective or a challenge. Therefore, evaluation of the current immunization schedule is not recommended if certain titer limits exist or a virus challenge is circulating in the field [67]. Because of several IBV serotypes, a single serotype of IBV vaccine does not provide more than 50% protection (Ciliary score ≥20), so it is not sufficient protection from a heterologous challenge on the basis of some clinical studies [16,68]. Higher ciliostasis protection (≥80%) was exposed through diverse vaccination of serotypes with Ma5 or H 120 at 1 day old, 793/B or 4/91 at 2 weeks against challenge at 5 weeks of age with two heterologous IBV is presented to be highly effective and good protecting the respiratory tract [16,69]. The results have revealed that applying two or more IBV live vaccines stimulated a higher expression of CD+, CD8, and IgA-bearing B cell in comparison to single serotype of IBV vaccine that promotes to a higher level of cellular and local immunity at the trachea against heterologous serotype [16,70]. Therefore, a vaccine should be selected from one of those serotypes to generate the broad protection of chicken. Alternative types of “multi/monovalent” vaccines, which have at least two serotypes of IBV vaccine or apply new serotypes depending on the high challenge in the farm and neighboring areas, are circulated via the same serotypes. Thus, virulent serotypes of IBV emerging in flocks can be identified by a serotyping assay, and confirmation of a particular vaccination program to protect against new IB isolates [16]. Several studies reported that IB vaccine with multiple serotypes can stimulate the proper antibody response and better protectives abilities against large number of different IBV types and novel emerging serotypes; especially local protection of the cilia in the trachea carried by the protectotype can inhibit secondary bacterial infections [41,71]. This program is generally used due to the more effective and cross-reaction immunity of birds based on actual practice and principle [19]. Virulent live vaccines are more effective and capable of reducing shedding the field challenge compared to recombinant vaccines, inactivated vaccines, and live vaccines, which are not highly homologous with a field challenge virus [72]. Most vaccines are inactive during the infective stage where birds are previously infected at the time of vaccination. As a result of acute infection, the immune organ is not fully working toward the activation of the immune system, as chickens do not generate a satisfactory level of antibody titer. Excessive rolling-type reactions are rarely found in close-contact houses, causing delayed immunity in the farm. The studied serology was conducted in 9 houses of chicken breeder flocks for the field trail (Tuaran breeder farm, Sabah, Malaysia), previously immunized with a few live and killed vaccines day 1 and day 18: Ma5 in spray; day 10: 4/91 in spray; day 42: IB killed; day 91 in Intramuscular injection: IB Mass in drinking; day 119: IB (4 in 1) killed in intramuscular injection; day 140: IB (4 in 1) killed in the pullet stage, and geometric mean titer (GMT) is shown for 0–30 weeks (Figure 4).

### 3.4. Group Size and Sample Number

Sample collection is important for the confirmation of diagnosis and identification during active and post-viral infection. In the acute infection of IBV, the majority of birds have a similar typical clinical sign; then, the number of samples can be reduced. At least five birds are enough to achieve higher confidence of the successful detection of infection by molecular assay [67]. In serology, more collection samples of birds having higher specific results are detected after infection; nevertheless, the costs and risk of false-positive results should be considered. In chronic infections in immunized birds, small quantities of the virus exist with lower prevalence, but require more birds for detecting infection. In the farm, a minimum of 20–25 bird samples should be taken from the group or every farm flock after vaccination, or results do not provide a good statistical data analysis. For the evaluation of vaccine efficiency, IBV antibodies were identified using a commercial ELISA kit according to manufacturer’s orders. Awad et al. [13] reported that serum was collected from days 3, 6, 10, 14, 18, and 25 post vaccination from 8 chicks/group to establish the mean or geometric antibody titer of an in vitro study. Therefore, statistical analysis is associated with bird welfare and financial reasons, and the choice of the number of chickens per group in experiments is relevant. Estimation of the required number of samples depends on studying the difference in levels of protection, the expected variability of the results between and within groups of chickens, and the desired confidence level [67,73]. A few researchers demonstrated sampling of the ciliostasis protection score system (CPSS) that was used to collect 10 tracheal organ cultures (TOCs) from 10 chickens per group using a five-class (scores 0–4) scoring system [74,75,76]. The principles of an official European publication reported that the efficiency of IB vaccines were required to use the binomial interpretation of CT on 10 TOCs from no less than 20 chickens per group, as used in many studies. Birds were protected, as no fewer than 9 of 10 rings presented normal ciliary activity [77].

### 3.5. Immunity at Time of Vaccination/Infection

The level of acquired immunity depends on many factors, especially at the time of vaccination or exposure of infection. Vaccines collaborate with the immune system to generate the same immune response as that in natural infection in birds. The humoral immune response has a significant role in the stimulation of certain levels of immune response compared with natural infection. Some researchers reported that vaccination is the process of immune development of the body originating from previous vaccination or earlier infection [78,79]. The humoral response normally increases after infection in vaccinated chickens, but could gradually decline or be delayed on the basis of current infection exposure in the farm. Therefore, the sensitivity of antibody tests could be much lower in vaccinated chickens compared to in unvaccinated chickens [48]. A study reported experimental IBV infections applied in vaccinated and unvaccinated birds, with results identifying a homologous virus challenge in much shorter and lower amounts in vaccinated than in unvaccinated chickens [80]. Different levels of humoral response could be identified by serology testing after 4 weeks of infection or 6 weeks after vaccination [81]. Neutralizing antibody (anti-IBV IgA) responses appear to follow the challenge, becoming more enhanced as time went by between vaccinations and challenge [82]. For instance, vaccines require a certain time to develop body immunity, subsequently carefully investigating the optimal period for applying primary and following booster immunization with the purpose of generating an optimal immune response in birds. The immunized are either too early or too late with secondary vaccinations, or it might fail to adequately strengthen the immune stimulation resulting in poor immune protection [83]. To prolong built-up immunity, primary immunization is followed by secondary immunization at an interval of 2 to 4 weeks; then, it may need an additional killed vaccine to develop complete immune protection [84], as shown in Figure 5A. Moreover, annual booster shots are needed to ensure that the immune system is still operative and to increase the boost of immunity. Although the early vaccination of IBV has a negative impact on circulating maternal antibodies until 4 weeks, local antibodies subsidize the local protection of respiratory epithelium against homologous IBV challenge. Therefore, primary vaccinations (live and killed) allow for overcoming the immunity gap, as they require a balance to generate the active immunity in a maternally derived antibody (MDA) period decline rather than in the late vaccination [85] (Figure 5B).

### 3.6. Age and Maternally Derived Antibodies

Maternal antibodies can provide passive immunity to chicks from hatching and temporarily protect against pathogens to the young immune system [86,87]. Age and maternally derived antibodies (MDA) are two significant reasons for the detection of antibodies during viral challenge or vaccination. However, high levels of MDAs in young chickens affect the multiplication and development of live vaccine virus, reducing the number of titers in serology. Studies suggested that only DNA vaccines (ovo- or intramuscularly) can provide the complete protection of day old chicks (DOC) due to vaccine antibodies, but they do not neutralize MDA [88]. Passive immunity has a comparatively short period, and MDA level was significantly increased after 3–4 days of age until invisible levels at 2–3 weeks of age [87,89]. Birds were more susceptible to infection at 2–3 weeks of age, while MDA protection declined, and the immune system of birds did not ever fully develop to respond against an early challenge [90]. The degree of immune response relies on age or MDA variation since humoral immune response gradually decreases if infection comes at a young age. The capacity of young chicks to develop their own antibodies after a response to antigenic processes is from hatching to older than 5–6 weeks of age. Antibody titers were considerably higher in specific-pathogen-free (SPF) chickens when immunized with S1, S2, and N proteins at 14 days of age than in those vaccinated at 1 or 7 days of age [91]. During that time, immunoglobulin (IgM) was not identified after the vaccination of DOC broilers with MDA, whereas IgM was noticeable after vaccination at 14 days of age. Some researchers worked on SPF chicks to apply live-attenuated IBV vaccines in a DOC hatchery receiving good local protection, which is currently applied in most broiler farms [92,93]. Moreover, some vaccination studies reported testing broilers after 2 weeks of age [94,95] and applying vaccination in 3 weeks [96] or even after 5 weeks of age [97]. Result comparison between variations of age is too difficult; nevertheless, the recurrent immunization of broiler chickens (one or two diverse variants) is now common practice in DOC followed by a boosted immunization at 2 weeks of age. The results of those studies revealed that the assessment value is more productive and protective against variant IBV challenges [18].

### 3.7. Genetics

Chicken breeding line is an important factor to genetic resistance to vaccination since variations can be genetically exposed and significantly influenced between inbred chicken lines that are infected with IBV and *E. coli*. Cook et al. [98] reported that there was no significant variation in serum antibody titers using VNT and ELISA after IB infection in experiments in highly resistant and highly susceptible chicken lines. However, another study showed that using inbred chicken lines challenges only IBV, indicating that the inbred line carrying the MHC B2 haplotype was more susceptible to IBV than line 15I (MHC B15 type) birds [99]. Joiner et al. [100] applied the same inbred chicken lines to show that the occurrence of clinical symptoms of an IBV challenge was considerably lower in a B2/B15 line than that in a B2/21 line. The MHC B15 haplotype was shown to be closely linked with resistance to IBV infection. However, no other study has been conducted to examine current commercial or SPF chicken breeds.

### 3.8. Vaccine Dosage

Vaccine doses are commonly monitored with regard to recommended manufacturer labels and instructions to deliver the correct volume to each chicken [18]. In a good immunization program, vaccine doses are a key factor in order for every bird to properly receive protection against IBV challenges [73]. Limited methods were established to determine the vaccine titer used in protection studies in embryonated eggs, calculated as the 50% egg infective dose (EID50). However, tracheal organ cultures (TOCs) were used to observe sensitivity in embryonated eggs for isolating IBV and to provide reproducible test results [101]. Moreover, titers are significantly varied depending on the expression in EID50 or median ciliostatic doses [62]. Alternatively, great variation was observed in different studies on the actual vaccine dose applied in many experiments. Another study suggested that vaccine dose is commonly applied at approximately log10 3.0 EID50/bird, but it is difficult to calculate the actual dose received by each chick [102,103]. Special attention is specified in the earlier preparation of vaccine suspension; the uses of a calibrated dropper and application directions have great impact on the actual vaccine dose received by each chick. It was determined that the dose of spray (hatchery or farm) and eye drop vaccine with IB Mass strain are followed with the manufacturer-recommended dose (1×) as ensued in 99%–100% of the birds positive for the vaccine virus at days 7–10 post vaccination, except the doses of ArkDPI vaccine strain tested in a hatchery spray cabinet with 100 times higher dose, which is not economically acceptable [104]. Moreover, spray vaccine is more preferred for IB virus, particularly while vaccinating for the early stage for local protection by adhering to the mucosa cells of the bird’s eyes and upper respiratory tract.

### 3.9. Vaccination Schedule

The timing of the IBV vaccination is given much attention due to the incidence of viruses circulating in certain farms being vaccinated. Broiler chickens are immunized 1 or 2 times against IBV due to their short lifespan [105]. Currently, DOC broiler chicks from hatcheries are mass-vaccinated with 1–3 serotypes of live-attenuated IBV vaccine for local protection before being shipped to the farm [106]. In several broiler farms, only a single vaccination is preferred for IBV due to shorter-lifetime broilers (≤35–45 days) [107]. In the case of longer-living broilers (≥49 days), a booster needs to be frequently applied between days 14 and 18 of age boost immunity duration. For IBV variants, vaccines can be administered (3.6 log10 EID50/dose) at day 1 for chicks and day 7 of older chickens by spray, intranasally or ocularly, or via drinking water. For breeding and laying birds, much longer cycles require both live-attenuated and killed vaccines starting from DOC followed by 18–20 weeks of pullet age for immunity longevity [108]. On the basis of the vaccine program, a live-attenuated IBV vaccine received by birds on the first day in a hatchery continues to 2–3 weeks of age on the farm, followed by 4 and 6 weeks of age. This vaccination schedule is properly standardized in the flock base for closely monitoring the serology status of vaccinated farms. Once birds start producing eggs, most killed IBV vaccines are given for longer immunity duration until the beginning of laying. Some farmers occasionally give a live-attenuated booster vaccine every 6 weeks in the laying stage if there is any challenge circulating in the farm area. It is common that killed IBV vaccine are combined with other killed vaccines, such as Newcastle disease virus (NDV), Reo virus, and IBD vaccines, to enhance layer variability and reduce stress during intramuscular injection.

### 3.10. Cross-Reactions between Serotypes

Diverse types of serotypes or IBV genotypes are globally documented, but different serotypes of the virus are not fully cross-protected. Despite this, novel, more infectious variants have emerged, but the amino acid sequence of the spike protein (S1) and genome are unchanged. Cross-reaction is particularly protected in chickens because of the interaction of multiple vaccinations covered by more than one included serotype [42,81]. Birds are challenged by new serotypes that differ from the serotypes of selection vaccines, so they are not fully protected against the prevalent serotype. On the basis of serological analysis, those heterologous cross-reactions in serotype-specific tests can be distinguished from homologous responses. Periodically, serological cross-reactivity is exposed to be higher in older birds depending on whether the birds had been previously contracted with infection or immunized many times with different serotypes. Few heterologous vaccines can decrease productivity and reduce the shedding of field viruses during the challenge of infection, initiating huge amounts of virus multiplication in the surroundings and leading to poor immune responses to vaccines. For this reason, serological data can be difficult to analyze in such conditions, which hinders the quality of diagnosis. A specific IB serotype can be recognized if the serotype is included in the group of tested IBV; otherwise, maximal cross-reaction should be undervalued.

### 3.11. Vaccine Handling and Storage

The supervision of vaccine handling, transportation, and storage greatly influences the efficacy and quality of vaccines. A live vaccine is sensitive to over 2–8 °C in storage temperature, so it should be continuously monitored to avoid over freezing until hardening [109,110,111]. Vaccines can spoil or lose superiority when in contact with extreme heating and intense light, leading to faster loss of strength, which greatly affects immunity development in birds (https://apps.who.int/iris/handle/10665/69387). A diluent is mixed before application, and temperature should be maintained by a cooler dropper during ocular and drinking live vaccinations. For IBV vaccinations, the process of live vaccination requires no more than 1–2 h of inoculation or drinking time after exposure to the vaccine [112]. In drinking-water vaccination, the water tank should be clean and containing ice cubes or chiller water until the temperature is maintained at below 20 °C. The actual amount of drinking water needs to be calculated the on previous day to avoid longer vaccination. For killed vaccines, a vaccine is thawed from storage around 24 h before vaccination, but should not exceed 37 °C for more than 5 h. Sunlight can directly destroy the vaccine, so it is prohibited to come into contact with direct sunlight, and the vigorous handling of bottles should be avoided [109]. To ensure vaccine quality, the broken emulsion layer in each batch of received killed vaccines needs to be observed before vaccination (Figure 6). Temperature-monitoring devices (TMDs) are used in recording intervals of at least 30 min that set up vaccine storage and transportation for monitoring the frequency of temperature readings, suitably maintained as required.

### 3.12. Postvaccine Challenge and Reaction

Signs of postvaccine reactions or challenges are characterized on the basis of local reaction, fever, and delayed reaction, and anaphylaxis is a common lesion or systemic reaction indicating that the chicken is reacting to a satisfactorily vaccination or good immunized response against IBV. These types of adverse postvaccine reactions are commonly found in infectious laryngotracheitis (ILT) rather than in IBV. A reaction is more severe in a few cases, with secondary infections such as conjunctivitis and swelling in the facial region, as shown in Figure 7. In most cases, birds react approximately 3–7 days after inoculation, showing slight respiratory symptoms of IBV [113]. Eventually, there are no clear symptoms in a successful vaccination, and blood samples should be collected for monitoring the actual titer in the serum for conformation.

### 3.13. Management Factors

The purpose of vaccines is for disease control and prevention against respective pathogens that are correlated with the limits of good biosecurity measures. However, if there is a sudden breakdown of the biosecurity program with a higher load of field challenge, a vaccination program does not help at a satisfactory level. Many management issues, such as congested spaces, uncontrolled temperature and humidity in the shed, poor nutrition, worm infestations, poor uniformity, and other secondary infections that reduce the bird’s immune system, can directly affect vaccination efficacy. Several studies suggested that continuous exposure to 30 ppm [114], 52 ppm [115] NH3 levels which are critical environmental stressors to laying and broiler chicken leading to reduced immunity as proven by decreased total plasma (IgA, IgG, and IgM) and complement concentrations. Furthermore, ammonia level in sheds being more than 70–100 ppm has adverse effects on chickens’ ability to yield local immunity, and birds become exposed to viral infections [116,117,118,119,120].

#### Ventilation Quality

The operation of ventilation systems is a vital part to regulate sufficient air exchange to meet the air quality in poultry houses to reduce disease susceptibility and stress. Poultry houses’ ventilation systems impact its ammonia levels, harmful concentrations of CO_2_, and hydrogen sulfide (H_2_S), which are continuously produced by chickens through defecation and urination [121,122,123]. The ventilation system with an exhaust fan is needed to provide a suitable environment in poultry houses by removing excess heat, moisture, dust, and odors [124]. A good vaccine response has been observed during the proper ventilation system, which is essential to maintain in the whole flock during vaccination except spray vaccine, fans should be off because of the uniform vaccine distribution. Poor quality ventilation can damage the lining of the respiratory tract through initiating the immune system and can cause chronic inflammation that leads to less (15% lower IgM response) efficiency or no response to vaccination. [109,125].

### 3.14. Possible Immunosuppression

Some immunosuppressive diseases and cyclosporine are the most common agents that directly influence IBV vaccination. Immunosuppressive agents are stimulated in early bird age to reduce the response of IBV vaccination. Thompson et al. [126] compared the humoral immune response after the IBV immunization of 5-week-old SPF chickens that were divided into two groups, vaccinated and unvaccinated, with IBD as day 1 chicks. The percentage of antibody responses after IBV inoculation was significantly higher in the vaccinated group than in the group given both viruses. Several viral diseases developed immunosuppression, such as infectious bursal disease (IBD) or Marek’s’ disease, inclusion body hepatitis (IBH), chicken infectious anemia (CAV), and mycotoxins, causing an increased chance of susceptibility to IBV infection [54,127]. Immunosuppression can reduce the activation or efficacy of the immune system, where the humoral and cell-mediated immune system does not work effectively.

### 3.15. Routes of IBV Vaccination

#### 3.15.1. Spray Vaccination

Spray vaccination is currently practiced in modern hatcheries at an early age of birds in farms, where a huge number of chicks require to be simultaneously vaccinated. The live vaccine of IBV is usually used for mass vaccinations in hatcheries or farms by fine/coarse spray via forcing through small spray nozzles, forming aerosolized droplets that chicks can easily inhale to uptake the vaccine [127,128,129], as shown in Figure 8A–C. Many factors are involved in correct spray vaccination (Table 1), such as droplet size with pressure, spray duration and constancy, maintained sanitation and hygiene, handling the equipment with a correct vaccine dosage, and the skills of workers, confirming the quality of vaccine delivery. A coarse rather than fine spray is more favorable to apply in DOC because fine droplets can be deeply inhaled in the respiratory tract, causing adverse reactions. The specific spray nozzle depends on the variety of sizes that form the measured droplet size. It was suggested that the minimal droplet size makes it harder for chicks to effortlessly inhale [130]. Several laboratory studies were conducted to measure the effectiveness of spray vaccination through electron microscopy, but no evidence was found for structural damage of the spike protein or the cracking of the virion membrane of the virus. Sometimes, the titer is superficially missed from the mechanical force applied to the vaccine during the handling process of vaccination [131]. Alternative studies observed a significant drop in vaccine response while comparing droplets from the spray nozzle to the absorbed level by the chicks, and proposed that the virus was being lost to the environment during the spray process [132,133]. Smaller volumes are used through a smaller nozzle flow rate, which helps the creation of smaller droplets upon aeroionization in spray vaccines. Generally, maximum hatchery coarse spray vaccination is accomplished with 100–150 µL droplets; however, this range is not persistent as it is just a normal of the total droplet sizes used [134,135].

#### 3.15.2. Oculonasal

The perfect route of IBV vaccination is normally evaluated through the oculonasal route against respiratory pathogens causing respiratory diseases [73], as shown in Figure 8D. However, this method is not suitable for application in commercial mass vaccination, and vaccines are much better stimulated in respiratory or conjunctival rather than parenteral routes [137]. It is assumed that immunization in the nasal or conjunctival route can avoid the neutralization effect of serum antibodies causing losses of epithelium lining. Chicks normally produce three classes of immunoglobulins, namely, IgA, IgG, and IgM or IgY, where IgM and IgA predominate in the serum and tears after local vaccination, and the IgA class prevails in most secretory channels. Huge amounts of plasma cells secreting antibodies against viral antigens are observed in the Harderian gland. Similarly, an increased number of antibodies in the serum and lacrimal fluid are associated with a reduction in viral concentration that can play a vital role in the stimulation of both systemic and local immunity [138]. Toro et al. [139] experimented on all routes of vaccination with cloacal route, resulting in higher identified IgG levels in the ocular route than those in drinking immunization. Experimental studies reported that several dilutions of the IBV vaccine (H120) immunized specific-pathogen-free (SPF) broilers by different vaccine routes, resulting in higher and rapid IBV-specific IgM response, at least 10 times the recommended dose (1000 times higher) given via drinking water [73].

#### 3.15.3. Drinking water

Most live IBV vaccines are applied in drinking water in broiler chickens 1 week following the booster dose if needed, depending on the farm situation. Broiler breeder and layer birds are commonly chosen for live vaccines in drinking at the pullet stage, and subsequently prime-boosted every 6 weeks in the laying stage, while the birds have lower titers or lower protection. In an attempt to mimic field conditions, drinking-water application is a good technique to avoid stress [140,141]; nevertheless, it is too difficult to quantify the actual dose received by each chick, and the consistency of vaccination within the group is impossible to determine. Monitoring should be performed to ensure that vaccination via drinking water is free from other medications, disinfectants, or chlorine at least 48 h before vaccination [142].

#### 3.15.4. Intramuscular Injection

Inactivated killed IBV vaccines are applied in the breast or leg muscle using intramuscular (IM) injections, as shown in Figure 8E. Killed vaccines should be maintained at ambient temperature (approximately 28 °C), properly shaken, and thawed before being applied in the field. If stored in a cool box in a dark site, an inactivated vaccine can restore its potency in 1–2 weeks outside a freezer. Killed vaccines are relatively more active when the birds receive earlier immunization with live vaccines, and revaccination is commonly advised every 6 weeks for prolonged immunity action [66,143]. The accidental injection or contamination injectors into the vaccinator of killed vaccines lead to serious local reactions, or muscle or skin inflammation [12]. A trained vaccine team should be knowledgeable during every shot of the vaccine in the breast muscle, maintaining a 45° angle, and the team should sanitize all vaccine instrument after vaccination.

#### 3.15.5. Gel Vaccination

A gel diluent is another of the latest practices for the application of the IBV vaccine. This technique was first used in the cocci vaccine in poultry as a delivery route of the vaccination process [144,145,146]. Due to gel thickness, it is created as a large droplet on which chicks obtain persistence steady until chicks are groomed by shank. The advantage of this technique is the increased ingestion of cocci oocysts that are currently used in cocci vaccine with live IBV, as shown in Figure 8F. This application system is currently trialed for IBV application for immunization against IBV [147,148]. However, laboratory study and commercial field trials revealed it to be more effective than spray vaccination [144,149,150]. It still needs additional trials or field research to review gel vaccination as a practical replacement of the spray route for IBV vaccines.

### 3.16. Vaccine Index (VI)

The success of vaccination performance is measured by mean titer response (mean antibody level) and percentages of coefficient of variation (CV) (value of measure’s uniformity) in vaccinated flocks. The overall standard for CV% uniformity after vaccination is measured as follows: less than 40%, excellent; 40%–60%, good vaccination; and more than 60%, vaccine methods need to be improved [151]. The ratio between mean titer and CV percentage is critical in assessing vaccine quality, defined as the vaccine index (VI), which is calculated as follows: (3)$$\mathrm{VI}=\frac{\;{\left(\mathrm{Mean}\;\mathrm{Titer}\right)}^{2}}{\mathrm{St}\;\mathrm{Dev}\;\times \;100}=\frac{\mathrm{Mean}\;\mathrm{Titer}}{\%\;\mathrm{CV}}$$

The VI score is estimated to provide a higher score (high mean titer, low CV percentage) in a good vaccination, and a low score with poor vaccination (low mean titer, high CV percentage). Depending on the standard parameters of different commercial kits.

## 4. Differential Diagnosis of Infectious Bronchitis Virus

The actual diagnosis of IBV infection is confusing with regard to clinical signs and postmortem lesions due to similar clinical signs produced by several respiratory pathogens, such as Newcastle disease (ND), avian influenza (AI), infectious laryngotracheitis (ILT) and infectious coryza (IC), and mycoplasma (Mg) [7]. Earlier reports stated that all respiratory pathogens can cause upper and lower respiratory infection, and significantly reduce egg production, requiring evidence from both field and laboratory results [152]. Likewise, typical signs of various poultry diseases are compared with other related diseases, for example, ND virus is involved with nervous symptoms, the bloody vomiting of ILT, facial swelling with lacrimation of IC, higher mortality in AI, and edema-swollen eyelids in MG. Egg-drop syndrome (EDS) is differentiated from IBV by shell quality and dropped production without infection. The whitish shell color is an important mark in ND and IBV infections, but watery albumin is found only in IBV infections [153]. The mortality of birds can be increased through the combined infection of IBV with ILT or coinfection of IBV with IC followed by high respiratory challenge and dyspnea [154,155]. However, APV can aggravate respiratory signs associated with IBV, but both viruses prefer coinfection together [156]. APV signs mainly fluctuate in egg production with drop hatchability, but egg-drop severity is found in IB infections. Few respiratory pathogens are associated with synergistic or mixed infection with IBV, MG, and *E. coli,* leading to the increased severity of infection [157,158]. There is no more evidence of dual infections with IBV and fowl cholera, and it is uncertain if there is more coinfection of IBV and CAV [159]. The synergistic role of IBV with interaction of mycoplasmosis is characterized by the combination of cold stress causing severe airsacculitis problems in the initial stages of laying [160,161]. Moreover, IBD can increased susceptibility to IBV and reduce IBV antibody levels [162].

## 5. Conclusions

Currently, a live-attenuated and inactivated vaccine is commonly used in IBV vaccination. However, IBV outbreaks in poultry farms usually persist due to a lack of actual serotypes or variants in vaccination programs, and vaccine failure, which is a significant problem in commercial farming. Laboratory diagnosis is a useful tool for the identification of actual serotypes or variants for vaccination programs to select accurate vaccines for the prevention and control of IBV. Moreover, we reviewed the best ways to protect from IBV through the successful vaccinations in every bird to maintain all necessary factors affecting the failure of vaccines. This review outlines this present situation of various vaccine applications, with correct approaches and factors that influence IBV diagnosis with differential diagnosis to assist in the accurate identification of IBV for necessary prophylactic measures to reduce economic losses from IBV outbreaks.

## Figures and Tables

**Figure 1 vetsci-08-00047-f001:**
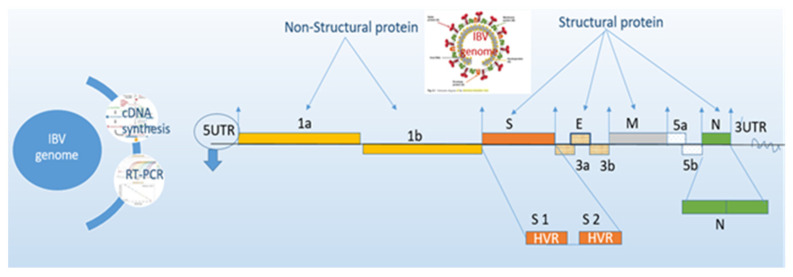
Morphology and genomic structure of infectious bronchitis virus (IBV) with different structural and non-structural genes.

**Figure 2 vetsci-08-00047-f002:**
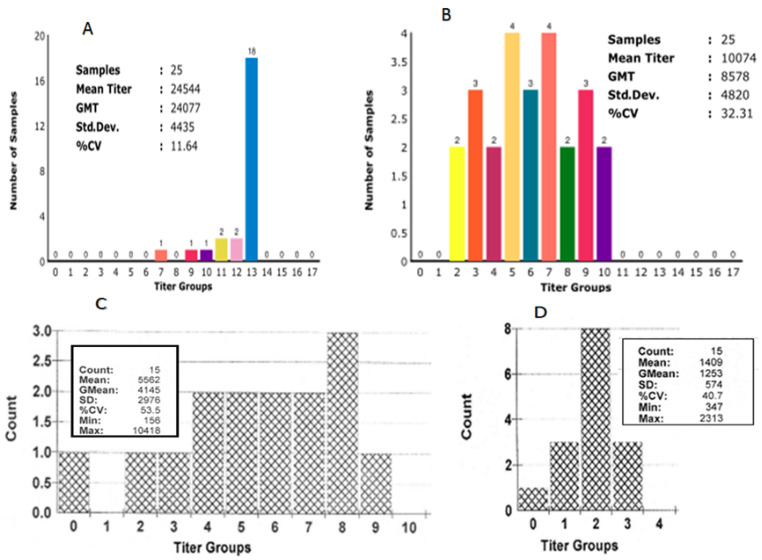
Compared serology analysis of IBV in two broiler breeder houses at 26 weeks of (**A**) challenged and (**B**) normal flock. Probable calculation of mean titer of challenged flock was three times more than that of the normal flock, and coefficient of variation (CV%) was too low compared with previous serology using Symbiotic test kit, USA. High-challenge titer after IBV infection in peak laying located almost in a single bar. In broilers, (**C**) high-challenge titer after IBV infection observed at the age of 35 days over 3–4 K titer calculated as post-IBV infection compared with (**D**) normal serology based on standard of Idexx ELISA test kit, USA. (Source: Personal communication; collected from Tuaran Broiler Farm, Sabah, Malaysia).

**Figure 3 vetsci-08-00047-f003:**
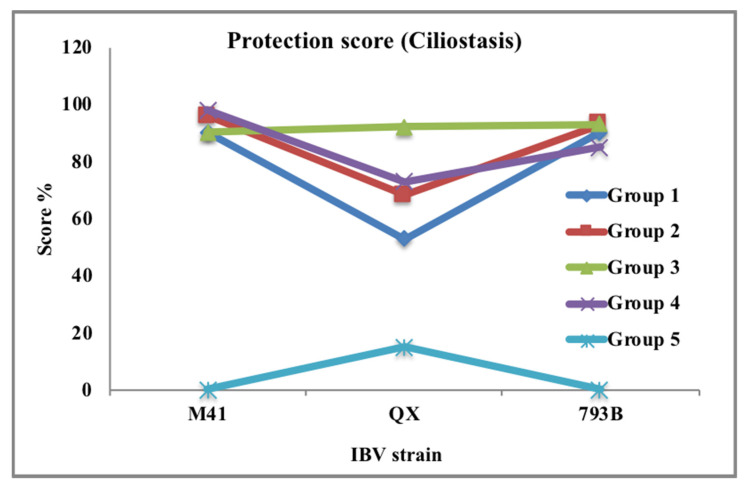
Ciliary protection stimulated by infectious bronchitis virus (IBV) vaccination programs against three virulent IBV strain challenges on day 21 post vaccine. IBV vaccination programs categorized into Group 1 = Mass1 + D274, Group 2 = Mass1 + 793B1, Group 3 = Mass2 + 793B2, Group 4 = Mass3 + Ark, and Group 5 = sterile water (SW). In this study, the monovalent Mass vaccines (H120 or Ma5) are referred to as Mass1 and Mass2, and the 793B vaccines are referred to as 793B1 and 793B2 (4/91 or CR88) serotypes. IBV challenged by M41, QX, or 793B initiated severe ciliostasis in unvaccinated–challenged birds. Ciliary scores indicated that vaccination programs gave good protection (>85%) against M41 and 793B. However, Group 3 (Mass2 + 793B2) was the only group to show good protection against QX; other groups were under partial protection. Protection score curve indicated that the higher the score was, the better the protection (Source: Awad et al. [66]).

**Figure 4 vetsci-08-00047-f004:**
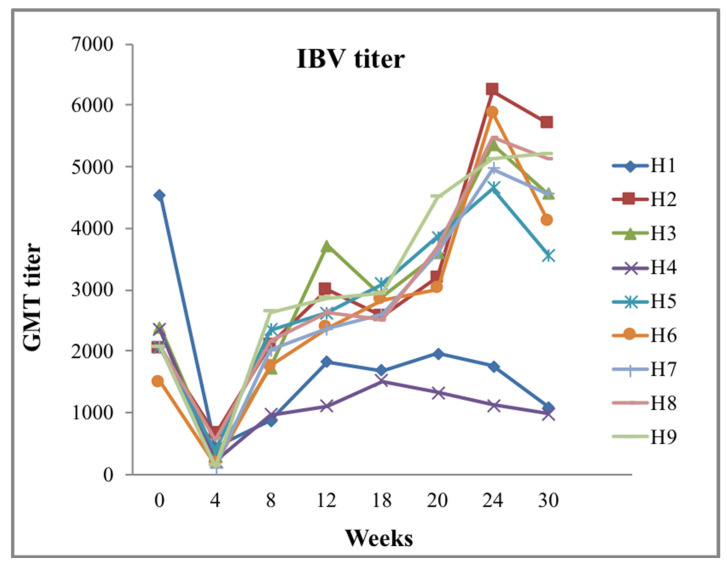
Geometric mean (IDEXX, USA) titer of nine broiler breeder houses in serology graph of 0–30 weeks. Birds vaccinated with infectious bronchitis (IB) live-attenuated vaccines followed by IBV variants and inactivated IBV-killed vaccination for 20 weeks during pullet stage. Out of 9 houses, House-1 and House-4 represent very low titer after vaccination compared with that of other houses. GMT titer was still below standard, indicating inappropriate vaccination with various involved factors during vaccination (X axis, weeks; Y axis, geometric mean titer (GMT)). (Source: personal communication; collected from Tuaran breeder farm, Sabah, Malaysia).

**Figure 5 vetsci-08-00047-f005:**
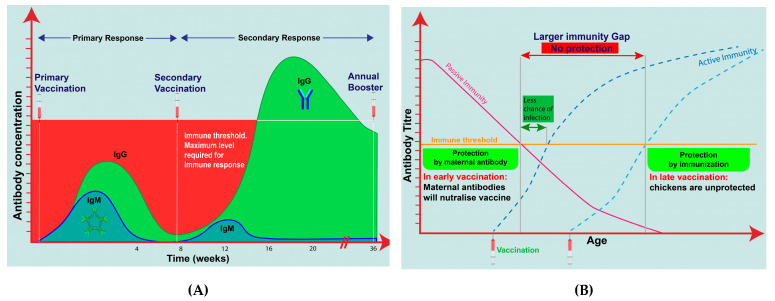
(**A**) Activity of primary and secondary immune response correlated to chicken vaccination with maintaining maximal level of immune response in full cycle of laying birds by using repeated annual boosting during decline of active immunity (https://microbiologynotes.com/differences-between-primary-and-secondary-immune-response/); (**B**) Assessment of immunity gap or no protection period between decline of passive immunity and start of active or acquired immunity causing birds to be unprotected and challenged, as active immunization is given protection in immunity gap (https://www.thepoultrysite.com/articles/protecting-the-immune-system-in-poultry-early-vaccination-with-vaxxitek-hvtibd).

**Figure 6 vetsci-08-00047-f006:**
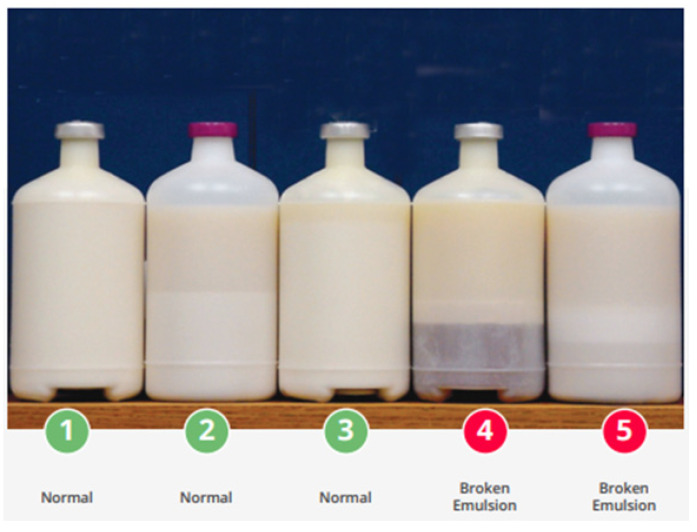
Broken emulsion shown in inactivated killed vaccine is caused by mishandling and storage issue (Source: www.cobb-vantress.com accessed on 28 December 2020).

**Figure 7 vetsci-08-00047-f007:**
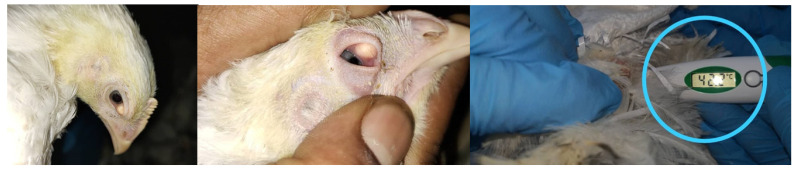
Sign of vaccine reactions or IBV postvaccine challenge in mildly infected eyes with slightly high temperature after 3 days post vaccination of IB Mass (day 21 and day 93) in field application.

**Figure 8 vetsci-08-00047-f008:**
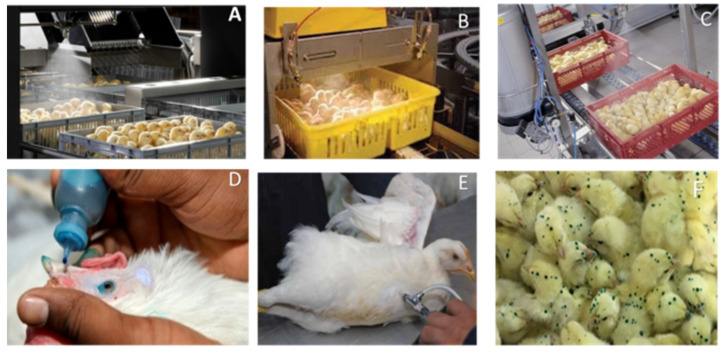
Different vaccination methods for respiratory infectious bronchitis virus. (**A**–**C**) Spray vaccine in hatchery; (**D**) oculonasal; (**E**) Intramuscular injection; (**F**) Gel-Pac vaccination of Cocci combined with IBV live vaccine (Sources: www.immucox.com, accessed on 29 December 2020).

**Table 1 vetsci-08-00047-t001:** Different types of spray vaccines with droplet size applied in poultry (source: www.wattagnet.com/articles/16484 [136]).

Types of Spray Vaccine	Droplet Size	Comments
ine spray	50–80 µL	More suitable for booster vaccinations on the farm; however, vaccine can penetrate into respiratory epithelial cells causing a post-vaccination reaction if applied too early in chicks.
Coarse spray	120–150 µL	Usually applied in day old chicks in hatcheries to administer respiratory vaccines and more convenient with high immune response.
Larger droplet sizes	<250 µL	Commonly applied to protect against Coccidia oocysts where chicks can easily gulp the droplet vaccine from their fluff.

## Data Availability

The study did not report any data.

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
