# Peer review of "Factor Influences for Diagnosis and Vaccination of Avian Infectious Bronchitis Virus (Gammacoronavirus) in Chickens"

_vetsci, 2021, doi:10.3390/vetsci8030047_

Round 1

Reviewer 1 Report

This review article highlights the factors that could affect effective vaccination for infectious bronchitis virus (IBV) in chickens and could affect appropriate diagnosis of IBV. The authors explain how diagnosis of IBV can affect vaccination strategies as well. Within the article, limitations for certain diagnostic factors or vaccination factors are presented. Influences of management, biosecurity, and nutrition are also mentioned for affecting IBV prevention. The authors describe the importance of preventing IBV among the commercial chicken farming industry. 

Major Revisions: 

  • Many sections do not cite references, and it is unclear which information is newly presented in this article.  For example, sections 2.11, 2.15.4, 2.15.5, 3.7, and 4 are some sections that lack citing references. 
  • The figures with graphs are difficult to read. The graphs should be consistent with formatting (color, axes types, titles, etc.). 
  • As the figures are described in the text about data with broiler breeders, it is unclear whether these figures presented are original (particularly figures 1, 2, 3, and 7). The data that are used in the figures should be cited in the text as well as in the caption. Naming the study rather than just adding the numbered reference is more helpful. If the figure is reproduced, there should be a statement describing allowance to do so. 
  • Within sections describing particular studies and specific data, there needs to be clearer indication as to which study is being referenced. Naming the author (et al.) with the date of the publication or a similar format would help delineate the particular study being referenced. Although some sections of this article use this format, many sentences (usually in the first half of the paper) have unclear reference to specific data. 
  • Describe where the data to produce Figure 2 originated and reference appropriately. 
  • Indicate whether this review article is intended for worldwide or general chicken production or for a particular region. If it is intended for worldwide use, indicate any regional differences in chicken production that could affect IBV. 

Minor Revisions: 

  • The headings and subsections make the document easier to follow; however, some subheadings could be re-named for clarity. 
  • Recommend changing “Poultry” in the main title to “Chickens.” 
  • Within many sections in this article, different populations are mentioned (such as chicks, broiler breeders, layers, etc.). The vaccination section should consistently specify which populations are being described. 
  • The ventilation quality should be mentioned within the management factors subsection. 
  • Add the word “response” after “immune” in line 50. 
  • Change the word “sabotaged” in line 53. 
  • Re-phrase sentence within lines 66-70. 
  • Consider changing the vaccination heading to indicate ‘effectiveness.’ 
  • The sentence within lines 82-84 should name the reference in the text rather than just citing the numbered reference since a quotation is used. 
  • It would be helpful to describe or specify the “control” stated in line 94. 
  • Rephrase sentence within lines 141-145. 
  • Change “mild infection” in line 362 to “mildly infected.” 
  • Line 396:  It seems that the word “than” is missing after the word “rather.” 
  • Figure 6:  There is no description of image F included in the caption. 
  • Rephrase section 2.15.3 subheading to “Drinking Water” or something similar to make it clearer. 
  • Line 470:  Define CV here, as it is first mentioned here. 
  • Consider changing the diagnosis heading to refer to accuracy of diagnosis. 
  • Rephrase sentence within lines 512-514. 
  • Change “qRT-PCR” in line 502 to “RT-qPCR.” 
  • Throughout the document, the phrasing/syntax could be more succinct and thus clearer. There are also many punctuation errors. Recommend having thorough proof-reading. 
  • There are some typographical errors in the reference list (check author name spellings, titles, formatting, etc.). 

Author Response

The revised manuscript has included several references in the text using yellow colour.

Regarding figures are provided the source of taken and increase the dpi

The authors would like to give a lot of thank to reviewer for spending more time to look through whole manuscript and giving several constructive comments for improving the quality manuscript.

Please see the attached response of your comments.

Reviewer 2 Report

Suggestions attached to the PDF send out to the editors. Various types of comments, forms and science. Important to check the iconography sources, and to make sure that they are not proprietary.

Author Response

The authors have revised the whole manuscript (yellow colour) according to reviewer constructive comments and suggestions.

Reviewer 3 Report

The manuscript is well written; however, I suggest that authors should improve the introduction regarding virus classification and schematic genome organization. Still, I suggest that the authors add some papers correlating genotyping and cross-protection among the different genotypes are circulating the world. For example:  the papers were published by J J Sjaak de Wit and co-authors.   

Author Response

The authors are highly appreciated reviewer suggestions and comments so the revised manuscript is provided all information.

Round 2

Reviewer 1 Report

The authors improved the previously submitted manuscript by adding specifications and clarifications for the reviewers’ comments and adding references to the subsections and figures.  There needs to be a clearer notation if figures are re-produced with permission (in addition to the citation).  There are still some syntax issues and typographical errors.   

  • Lines 35-36:  Rephrase “under” -- are you trying to express that betacoronaviruses include human coronaviruses such as SARS-CoV? 
  • Figure 1:  Was this figure re-produced or original?  Please cite. 
  • Line 76:  Add the word “response” after “immune.” 
  • Line 77:  Change back “immune response” to “immunity.” 
  • Line 79:  Rephrase – are you trying to express that farmers are blamed for vaccination failure or that farmers blame the vaccine’s lack of effectiveness for failure to immunize their flocks? 
  • Line 95:  Delete “, they.” 
  • Please add a reference for lines 135-137. 
  • Please add a reference for lines 140-141. 
  • Line 155:  Change to “than the expected.” 
  • Line 173:  Typographical error for titers. 
  • Line 178:  Add “to” before “more than.”  Delete the “s” on “serotypes.” 
  • Line 266:  Add a period. 
  • Line 272:  Typographical error for confer. 
  • Line 319:  Change “compare” to “comparison.” 
  • Lines 327-330:  Rephrase. The sentence structure is difficult to follow the authors’ point. 
  • Line 340:  Typographical error for field. 
  • Figures 5, 6, and 8:  Were these figures granted permission to re-produce from the original source? 
  • Line 481:  The last part of the sentence is unclear.  Should “that” be changed to “which?” 
  • Line 562:  Change to “mildly infected.” 
  • Line 633, figure 8F:  There still is a description lacking for part F of figure 8. 

Author Response

To

Chief Editor

Veterinary Sciences

Ref. MS. No. Vetsci-1021208

Manuscript Title; Factor Influences for Diagnosis and Vaccination of Avian Infectious Bronchitis Virus (Gammacoronavirus) in Chickens

Dear Editor,

The authors pleased to inform you that we have revised the above manuscript according to the Editor and Reviewer’s suggestions and their valuable comments. We have made some necessary corrections in the text as to the comments and suggestions given throughout the editors and reviewers comments. These are listed below point to point:

Comments of Reviewer 1:

This review article highlights the factors that could affect effective vaccination for infectious bronchitis virus (IBV) in chickens and could affect appropriate diagnosis of IBV. The authors explain how diagnosis of IBV can affect vaccination strategies as well. Within the article, limitations for certain diagnostic factors or vaccination factors are presented. Influences of management, biosecurity, and nutrition are also mentioned for affecting IBV prevention. The authors describe the importance of preventing IBV among the commercial chicken farming industry. 

 Major Revisions: 

  1. Lines 35-36:  Rephrase “under” -- are you trying to express that betacoronaviruses include human coronaviruses such as SARS-CoV? 
  • Page 1, Authors have revised the sentence using yellow color

  1. Figure 1:  Was this figure re-produced or original?  Please cite.
  • The original figure is provided

  1. Line 76:  Add the word “response” after “immune.” Line 77:  Change back “immune response” to “immunity.”
  • Edited page 2 using yellow color
  1. Line 79:  Rephrase – are you trying to express that farmers are blamed for vaccination failure or that farmers blame the vaccine’s lack of effectiveness for failure to immunize their flocks?

  • Page 2, last paragraph, already refreshed the sentence.

  1. Lines 327-330:  Rephrase. The sentence structure is difficult to follow the authors’ point.

  • Page 9, first paragraph the authors have rephrased the sentence.

  1. Typographical error several pages mentioned by reviewers

  • The authors have provided corrected word in several pages using yellow color.

Thanks for your realistic comments and suggestions on our manuscript, and also your kind cooperation and assistances.

With best regards

Siddiquee, S

Biotechnology Research Institute

Universiti Malaysia Sabah

Reviewer 2 Report

Version 2 with major revision accepted

Author Response

(The authors gave the same response as above.)
